# Phytochemical Composition and Antioxidant Activity of Traditional Plant Extracts with Biocidal Effects and Soil-Enhancing Potential

**DOI:** 10.3390/antiox14101198

**Published:** 2025-10-02

**Authors:** Camelia Hodoșan, Cerasela Elena Gîrd, Ștefan-Claudiu Marin, Alexandru Mihalache, Emanuela-Alice Luță, Elena-Iuliana Ioniță, Andrei Biță, Ştefania Gheorghe, Laura Feodorov, Violeta Popovici, Elena Pogurschi, Lucica Nistor, Iulius Sorin Bărbuică, Lăcrămioara Popa

**Affiliations:** 1Formative Science in Animal Breeding and Food Industry Department 59, Faculty of Engineering and Animal Production, University of Agronomic Sciences and Veterinary Medicine, Mărăsti Bvd. District 1, 011464 Bucharest, Romania; 2Faculty of Pharmacy, University of Medicine and Pharmacy “Carol Davila”, Traian Vuia 6, 020956 Bucharest, Romaniaemanuela.luta@umfcd.ro (E.-A.L.); lacramioara.popa@umfcd.ro (L.P.); 3Department of Pharmacognosy & Phytotherapy, Faculty of Pharmacy, University of Medicine and Pharmacy of Craiova, Petru Rareș 2, 200349 Craiova, Romania; 4Control Pollution Department, National Research and Development Institute for Industrial Ecology ECOIND, 57-73, Drumul Podu Dambovitei Str., 060652 Bucharest, Romania; 5Center for Mountain Economics, “Costin C. Kiriţescu” National Institute of Economic Research, Romanian Academy, 725700 Vatra-Dornei, Romania; 6Innovative Therapeutic Structures Research and Development Centre (InnoTher), “Carol Davila” University of Medicine and Pharmacy, 020956 Bucharest, Romania

**Keywords:** plant extracts, natural antioxidants, polyphenols, phytochemical profiling, sustainable agriculture, biocidal activity, soil enhancer, ethnobotanical knowledge, *Daphnia magna* assay

## Abstract

This research provides a comprehensive evaluation of the phytochemical composition, antioxidant potential, and biological properties of four plant species with longstanding use in ethnobotanical traditions: *Calendula officinalis*, *Mentha* × *piperita*, *Urtica dioica*, and *Juglans regia*. Plant extracts were obtained using a range of solvent systems and subsequently analyzed for their content of total polyphenols, flavonoids, and phenolic acids. Ultra-high-performance liquid chromatography coupled with mass spectrometry (UHPLC-MS) enabled the accurate identification and quantification of major polyphenolic constituents. The antioxidant capacity was assessed through a series of in vitro assays, and elemental analysis was conducted to determine microelement content. To evaluate potential ecological implications, acute toxicity was tested using *Daphnia magna*, while phytotoxic effects were also examined. The results demonstrate pronounced antioxidant activity along with notable biocidal and soil-enhancing properties. These findings underscore the potential of such plant-based formulations as sustainable alternatives to conventional agrochemicals and highlight the relevance of integrating traditional botanical knowledge with modern strategies for enhancing soil quality, crop performance, and environmental sustainability.

## 1. Introduction

Healthy soil is the foundation of healthy food, and phytoremediation with plants offers a sustainable way to rebuild this foundation where soil has been degraded. By selecting appropriate species (e.g., plants with high biomass or capable of stabilizing/extracting metals) and associating them with beneficial microorganisms, pollutant loads can be reduced, resulting in more resilient crops and safer, more nutritious food with fewer harmful residues. The use of ethnobotanical heritage can form the basis of sustainable phytoremediation of the soil.

Ethnobotanical practices in Romania are deeply rooted in the country’s rich ecological and cultural heritage, reflecting a wealth of traditional knowledge on the use of plants for medicinal and agricultural purposes [1,2]. In rural areas of Muntenia (Romania), traditional practices involve the use of herbal macerates and extracts as natural pest control solutions in agriculture. These bio-mixtures were mainly prepared from *Calendula officinalis*, *Mentha* × *piperita*, *Urtica dioica*, and *Juglans regia*, with additional plants such as oak leaves, horseradish, wormwood, and elderberry leaves occasionally included, depending on specific needs. The effectiveness of these mixtures comes from the nutritional compounds and natural biocides found in these plants—such as polyphenols, flavones, phenyl carboxylic acids, volatile and nonvolatile terpene derivatives, and various reactive radicals—which have acted as deterrents against insects and other pests, providing a sustainable alternative to synthetic pesticides.

The species *Calendula officinalis* L. (marigold) has a significant ethnobotanical profile in Romania, being widely used in both traditional medicine and sustainable agriculture [3]. In agriculture, marigold flowers play a dual role, not only inhibiting pathogenic fungi such as *Fusarium spp*. and *Rhizoctonia solani* but also contributing to sustainable agriculture through its endophytic bacteria and phytocomplex (polyphenols, volatile and non-volatile terpene derivatives) that promote soil health and suppress phytopathogens [4,5,6].

*Mentha* × *piperita* L. (peppermint) is a plant highly valued in traditional medicine and sustainable agriculture due to its rich bioactive profile and therapeutic properties. The essential oil contains menthol, menthone, and isomentone, which have strong antimicrobial, antioxidant, and biocidal activities, making it effective against pathogens such as *Staphylococcus aureus* and *Escherichia coli*, as well as pests such as *Culex quinquefasciatus* and *Phthorimaea absoluta* [7,8,9,10,11]. Polyphenols contribute to its free radical scavenging capacity and enhance its antimicrobial effects [7,10]. In agriculture, peppermint essential oil has demonstrated its potential as a natural pesticide and herbicide, with applications in weed control and crop protection [12,13].

*Urtica dioica* L., commonly known as stinging nettle, is a perennial plant with significant ethnobotanical importance, valued for both its therapeutic and agricultural applications. Nettle is an ecological alternative to synthetic pesticides and fertilizers, with nettle slurry being used to support plant growth, enhance resistance to pathogens, and act as a natural pest repellent, making it invaluable in organic farming practices [14,15].

The species *Juglans regia* L, commonly known as walnut, is a versatile tree widely valued for its ethnobotanical, medicinal, and agricultural applications. In agriculture, walnut leaves and by-products such as husks and green husks offer environmentally friendly alternatives to synthetic pesticides and fertilizers. These by-products act as natural fungicides against pathogens such as *Fusarium culmorum* and *Botrytis cinerea*, supporting their use in sustainable agricultural practices. In addition, walnut shells are used as bioabsorbents to remove hazardous materials from wastewater [16,17].

Based on these premises, this paper investigates the phytochemical composition and antioxidant activity of plant extracts traditionally used for their biocidal effects and soil improvement potential, in the context of the transition to sustainable agricultural practices.

## 2. Materials and Methods

The rationale underlying the application of analytical methods is based on the need to obtain a complete and rigorous characterization of the phytochemical composition and biological activities of the extracts studied. First, spectrophotometric methods for the determination of flavonoids, phenolic acids, and total polyphenols were used due to their sensitivity and reproducibility, providing an overall assessment of the content of phenolic compounds, which are recognized for their antioxidant and biocidal role. Subsequently, UHPLC-MS analysis was applied to individually identify and quantify polyphenols, complementing the general information obtained by spectrophotometry and ensuring high accuracy in the profiling of bioactive compounds. The determination of microelements was included to correlate phytochemical value with soil improvement potential, since minerals such as Fe, Mg, and Zn play an essential role in plant metabolism and soil fertility.

An evaluation of antioxidant activity using standardized methods (ABTS, DPPH, and FRAP) was necessary to confirm the extracts’ ability to neutralize free radicals and highlight their applicability as natural sources of antioxidants. Finally, toxicity tests on *Daphnia magna* and phytotoxicity tests on seeds were included to verify the ecological safety of the extracts, ensuring that they can be proposed as sustainable alternatives to conventional agrochemicals without causing major negative effects on the environment.

### 2.1. Chemicals and Reagents

All solvents and reagents used were of analytical grade, purchased from Sigma-Aldrich, Schnelldorf, Germany. For the quantitative analysis and the radical scavenging activity, high-purity rutoside, tannic acid, chlorogenic acid, 2,2′-azino-bis(3-ethylbenzothiazoline-6-sulfonic) acid (ABTS), and trolox were used. For the extraction methods, analytical-grade methanol (99.9%) was used and diluted to specific concentrations with purified water. The solvents and the external standards used for the in-depth UHPLC-MS analysis were of analytical grade purchased from Sigma-Aldrich, Schnelldorf, Germany.

### 2.2. Plant Material and Extraction Process

The plant material consisted of dried aerial parts of nettle and mint, marigold flowers, and walnut leaves from the 2024 harvest. These were provided in their entirety by the Plant Genetic Resources Bank for Vegetable, Flower, Aromatic, and Medicinal Plants (BRGV) in Buzău (southeastern Romania, with approximate coordinates of 45°08′22.92″ north latitude and 26°50′07.80″ east longitude). According to a local ethnobotanical technique used for agricultural bio-treatments, these four species are part of the mixture, and the following extractions were carried out: classical infusion extraction, maceration, reflux extraction, and reflux extraction to dry extract, methods which are going to be discussed further.

In addition, these plant parts are among the most commonly used and easily accessible to people in rural areas, where traditional practices are preserved and passed down through generations. In Muntenia’s rural regions, the species examined in this study are frequently found in farmers’ orchards, making them highly relevant for local agricultural applications. Specifically, the combination of marigold and walnut is often included in traditional recipes for mixtures and macerates used in Romanian farming. Moreover, several studies indicate that co-extraction processes may enhance extraction efficiency and promote additive or synergistic interactions between phytochemical compounds, thereby increasing their overall bioactivity [18,19,20].

Different extraction methods (infusion, maceration, reflux and dry extraction) were used to cover a wide range of bioactive compounds and to evaluate the impact of the solvent and technique on the final yield. Infusion and maceration were chosen to replicate traditional ethnobotanical practices and to obtain mild extracts representative of popular uses, with only specific plants being used, each adapted to the type of plant material. Reflux with methanol of different concentrations allowed for the efficient extraction of polyphenols and flavonoids, and the comparison of solutions with 50% and 70% methanol highlighted the influence of solvent concentration on phytochemical content. Extraction with 99% alcohol was also attempted on Urtica dioica, but the solution obtained contained a high amount of chlorophyll, which made the analysis inconclusive, and therefore this method was not continued. The transformation of liquid extracts into dry extracts by evaporation and lyophilization ensured a stable and concentrated form, essential for detailed analysis and potential practical application. The choice of these methods is thus justified by the integration of tradition with scientific rigor and by obtaining results relevant to the phytochemical characterization and antioxidant activity of the plants studied.

For the infusion (MEI), approximately 5 g of powdered plant material (*M.* × *piperita)* were extracted with 100 mL of boiling, purified water for 10 min. After extraction, the mixture was immediately cooled and filtered, and quantitative assessments were carried out. The maceration (CJM) process was carried out as follows: approximately 42 g of mixed powder plant material (*C. officinalis* flowers and *J. regia* leaves in a 13:1 *w*/*w* ratio) were mixed with 1000 mL of purified water and placed in a cool place (between 5 and 8 °C) for one week. This mixture was stirred twice a day until the maceration process was finalized, following the quantitative assessment of the extract.

The reflux extraction was performed for each plant material (*U. dioica*, *M.* × *piperita,* and the mixture of *C. officinalis* flowers and *J. regia* leaves in proportions of 13:1 *w*/*w*). Accordingly, 5 g of powdered plant material was mixed with 100 mL of solvent (diluted methanol 50% and diluted methanol 70%) (*U. dioica* methanol 50% extract—UD50, *U. dioica* methanol 70% extract—UD70, *M.* × *piperita* methanol 50% extract—ME50, *M.* × *piperita* methanol 70% extract—ME70, the mixture of *C. officinalis* flowers and *J. regia* leaves methanol 50% extract—CJ50, the mix of *C. officinalis* flowers and *J. regia* leaves methanol 70% extract—CJ70) each of the six extractive solutions has undergone a reflux treatment for 30 min at 80 °C. After cooling the solutions were filtered and subject to the qualitative analysis.

Similarly to the extraction process described above, extractive solutions of the four plant materials (*U. dioica* dry extract—UDSE, *M.* × *piperita* dry extract—MESE, *C. officinais* dry extract—COSE and the mixture of *C. officinalis* flowers and *J. regia* leaves dry extract—CJSE) were obtained using diluted methanol (50%), concentrated in a rotary evaporator (Vacuum Pump V-700, Buchi Labortechnik, Flawil, Switzerland), and lyophilized (ALPHA 1–2 LDplus freeze dryer, Martin Christ Gefriertrocknungsanlagen GmbH, Osterode am Harz, Germany), thus resulting the dry extracts. The dry extracts were stored in sealed containers with Silica gel until use.

To be noted that, in the toxicity analysis, both individual extracts of *Calendula officinalis* flowers and *Juglans regia* leaves were evaluated in order to compare the specific effects of each species with those generated by the interaction of their bioactive compounds. This approach was necessary to identify any cumulative or antagonistic effects on test organisms, providing a more complete picture of the safety of using mixed extracts in agricultural and ecological applications.

### 2.3. Total Polyphenol Content (TPC)

The total polyphenol content was determined using a colorimetric method with the Folin–Ciocalteu reagent, similar to that described by Luță et al. [21], with slight modifications. For some of the extractive solutions, different dilutions were prepared due to the high concentration of active phytochemicals. From these dilutions, different volumes between 0.2 and 0.9 mL were diluted to 1 mL with purified water. Afterward, 1 mL of Folin–Ciocalteu reagent was added. This mixture was set aside for 5–8 min at room temperature. Afterwards, 8 mL of a sodium carbonate solution (0.2 g/mL) was added, and the final solution was left to stand for 40 min at room temperature in the dark. The absorbance of each sample was measured at λ = 725 nm with a spectrophotometer (Jasco V-530, Jasco Corporation, Tokyo, Japan), and it was recorded against a blank solution consisting of 1 mL of Folin-Ciocâlteu reagent, 1 mL purified water, and 8 mL of sodium carbonate solution (0.2 g/mL).

For the dry extracts of each plant material, approximately 0.1 g of extract was dissolved in a 50 mL volumetric flask, which contained solvent (methanol 50%), and was subjected to an ultrasound bath for 15–20 min. Finally, the TPC was determined as above, using an identical technique.

Simultaneously, a standard calibration curve of tannic acid was generated, using known concentrations of the reference substance ranging from 2.04 to 9.18 μg/mL, with the same protocol as above (R^2^ = 0.9990). The TPC was measured as mg tannic acid equivalents/g of plant material.

### 2.4. Total Flavonoid Content (TFC)

The TFC was established using a colorimetric method with AlCl3, which reacts with flavonoids to form a yellow complex. As mentioned above, some of the extractive solutions were diluted due to the high concentration of phytochemical components. Volumes between 0.2 and 1.6 mL were placed into 10 mL volumetric flasks, then 2 mL of sodium acetate (0.1 g/mL) and 1 mL of aluminum chloride solution (0.025 g/mL) were added to each sample, followed by adjusting to 10 mL with each different solvent for each different sample. At the same time, for each sample, a control was prepared by mixing the same volumes of the extractive solution used for analysis with each of the solvents in a 10 mL volumetric flask, essentially creating the samples from the extractive solutions without adding any reagents. After leaving the samples at room temperature for 45 min, the absorbance was measured at λ = 427 nm against the control sample with a spectrophotometer (Jasco V-530, Jasco Corporation, Tokyo, Japan).

In the case of the TFC for the dry extracts, as mentioned above for the TPC, different solutions were created from the 0.1 g of dry extracts; however, no dilution was needed in this case. After the ultrasound bath, the TFC assessment was carried out identically as the extractive solutions.

A calibration curve was traced using known concentrations of rutoside (5–35 μg/mL), utilizing the same colorimetric method as for the samples, with an R^2^ value of R^2^ = 0.9992. The TFC was measured as mg Rutoside equivalents/g of plant material [22].

### 2.5. Total Phenolic Acids (TAC)

Regarding the TAC evaluation, the chosen colorimetric method was based on the reaction of phenyl carboxylic acids with the Arnow reagent, which imparts a reddish tint to the solution. Due to the reaction’s high instability, the TAC was evaluated only for some of the extracts (for MESE, UD50, UD70, CJ50, and CJ70). For the liquid extracts, between 0.800 and 1.800 mL of extractive solution was added to 10 mL volumetric flasks. The following order of reagents was then added: 2 mL of 0.5 M hydrochloric acid, 2 mL of Arnow reagent, and 2 mL of 0.085 g/mL sodium hydroxide. Finally, the flasks were adjusted to 10 mL with purified water, and immediately the absorbance values were registered using the spectrophotometer (Jasco V-530, Jasco Corporation, Tokyo, Japan) at λ = 525 nm, against a control sample of each research sample without the Arnow reagent added.

As for the dry extract, 0.1 g of extract was dissolved in a 50 mL flask with 50% afterwards it was then subjected to an ultrasonic bath for 15–20 min, and the following procedure was carried out identically to the one for the liquid extractive solutions.

For this determination, a standard calibration curve was traced using known concentrations of chlorogenic acid (11.30–52.80 μg/mL) applying the same procedure as mentioned above (R^2^ = 0.9998). The TAC was measured as μg Chlorogenic acid equivalents/g of plant material [22].

### 2.6. Radical Scavenging Activity—ABTS Method

The ABTS method is one of the most widely used methods for measuring radical scavenging activity due to its sensitivity and accuracy. This evaluation was conducted similarly to Luță et al. [23]. The basis of this assay is centered around the free radical ABTS^•+^, which is the direct result of the reaction between a solution of 7.4 mM ABTS and another solution of 2.6 mM potassium persulfate. This mixture was prepared 16 h before the actual assay and stored at room temperature in the dark.

This assessment was carried out only for the dry extract solutions, which were prepared by dissolving 0.100 g of dry extract in 100 mL of 50% methanol. Ten volumes (0.100–1.00 mL) of the dry extract solutions were sampled into 10 mL volumetric flasks and diluted to 10 mL with methanol (50%). At least 10 dilutions were made of each dry extract solution. Formulated dilutions: 0.5 mL of each was mixed with 3 mL of the free radical ABTS^•+^ solution, which had been previously prepared. These samples were stored in darkness for 6 min, after which the absorbance was measured at λ = 734 nm against analytical-grade ethanol (99.60%) using a spectrophotometer (Jasco V-530, Jasco Corporation, Tokyo, Japan).

By applying the following formula, the ABTS^•+^ inhibition percentage was calculated:$$ABTS\;inhibition\;\%\;(IC)=\;\frac{Abs\;blank\;-\;Abs\;sample}{Abs\;blank}\;\times \;100$$
where

Abs blank—the absorbance of the free radical ABTS^•+^ solution.

Abs sample—the absorbance of each sample after 6 min in contact with the free radical ABTS^•+^ solution.

The IC50 value (the concentration of sample needed to scavenge at least 50% of the free radical ABTS^•+^) was determined by plotting a graph using the concentrations of dry extracts of the samples and the ABTS inhibition % (IC). The antioxidant activity of an extract decreases as the IC50 value increases. Additionally, a calibration curve was generated using known concentrations of trolox (0.01–0.10 mg/mL), yielding a correlation coefficient of R^2^ = 0.9926.

### 2.7. Diphenyl-1-Picrylhydrazyl Free Radical Scavenging Assay (DPPH)

Equal portions of the dry extracts were dissolved in 100 mL of 50% ethanol. From each resulting solution, ten aliquots of equal volume were transferred into 10 mL volumetric flasks and adjusted to volume with the same solvent. A 0.5 mL sample of each diluted solution was then mixed with 3 mL of 0.1 mM DPPH radical solution (Sigma–Aldrich, Hamburg, Germany) [23]. The mixtures were kept in the dark for 30 min, after which their absorbance was measured at 515 nm using a spectrophotometer (Jasco, Hachioji, Japan). Ascorbic acid (Sigma–Aldrich, Hamburg, Germany) served as the reference standard for the calibration curve, prepared in the concentration range of 2–22 µg/mL.

By applying the following formula, the DPPH inhibition percentage was calculated:$$DPPH\;inhibition\;\%\;(IC)=\;\frac{Abs\;blank\;-\;Abs\;sample}{Abs\;blank}\times \;100$$

Abs blank—the absorbance of the DPPH solution.

Abs sample—the absorbance of each sample after 6 min in contact with the free DPPH solution.

### 2.8. Ferric-Reducing Antioxidant Power Assay (FRAP)

The FRAP assay evaluates antioxidant potential based on the reduction in ferric ions (Fe^3+^) to ferrous ions (Fe^2+^) by antioxidants present in the samples. The development of a blue coloration indicates the formation of ferrous ions.

For each plant extract, about 0.1 g of dry material was dissolved in 100 mL of 50% ethanol. From each resulting stock solution, eight equivalent volumes were transferred into volumetric flasks and diluted to 10 mL with the same solvent. Subsequently, 2.5 mL of each diluted solution was mixed with 2.5 mL phosphate buffer (pH 6.6, Sigma–Aldrich, Hamburg, Germany) and 2.5 mL of 1% K_3_[Fe (CN)_6_] (Sigma–Aldrich, Hamburg, Germany), followed by incubation at 50 °C for 20 min. After heating, 2.5 mL trichloroacetic acid (Sigma–Aldrich, Hamburg, Germany) was added to each mixture. Then, 2.5 mL of distilled water and 0.5 mL of 0.1% FeCl_3_ (Sigma–Aldrich, Hamburg, Germany) were added to the resulting supernatant solutions. The samples were left to stand for 10 min, after which the absorbance at 700 nm was recorded against a blank prepared from 5 mL of distilled water mixed with 0.5 mL of 0.1% FeCl_3_. The antioxidant capacity was calculated using the EC50.

### 2.9. Microelement Assay

The method is based on dissolving a plant tissue sample through microwave-assisted disaggregation. The resulting solution is then atomized into the air-acetylene flame of an atomic absorption spectrophotometer, where the absorption of radiation at specific wavelengths is measured photometrically.

Each 0.5 ± 0.0002 g sample was accurately weighed and transferred into Teflon DAP–60 K disaggregation vials (60 mL capacity), using 7 mL of a reagent mixture composed of 65% HNO_3_ and 30% H_2_O_2_ (5:2, *v*/*v*). The vials were securely sealed with safety caps, placed in the rotor, and subjected to microwave-assisted disaggregation with the Microwave Disaggregation System BERGHOF, Speedwave MWS-2 Comfort (Eningen, Germany). The process followed a controlled temperature and power sequence: 8 min at 130 °C, followed by 5 min at 155 °C, and finally 12 min at 170 °C, all conducted at 80% of the microwave oven’s maximum power of 1000 W. Once the digestion was complete, the solutions were allowed to cool to room temperature, filtered through filter paper, and quantitatively transferred into 50 mL volumetric flasks, which were then brought to volume with deionized water. A blank sample was prepared under identical conditions. The final sample solution was atomized into the air-acetylene flame of the atomic absorption spectrophotometer (Thermo Electron—SOLAAR M6 Dual Zeeman Comfort—Cambridge, UK, with a deuterium lamp for background correction and air-acetylene flame or Zeeman background correction for graphite furnace), where absorbance was measured at specific wavelengths to determine the concentration of target elements.

### 2.10. In-Depth UHPLC-MS Polyphenol Assay

The analysis of phenolic acids was performed using an ultra-high-performance liquid chromatography (UHPLC) system, specifically a Waters Acquity Arc equipped with both a photodiode array (PDA) detector and a QDa mass detector. Chromatographic separation was carried out on a CORTECS C18 column (4.6 × 50 mm, 2.7 µm particle size) maintained at a constant temperature of 30 °C.

The mobile phase employed a gradient elution using two solvents: water with 0.01% formic acid (Solvent A) and acetonitrile with 0.01% formic acid (Solvent B). The elution started at a flow rate of 0.8 mL/min with 99% Solvent A for the first minute. Over the next 12 min (from 1 to 13 min.), the proportion of Solvent A was linearly decreased to 70%. This composition was held until 13.10 min, after which it was linearly decreased to 20% A between 13.10 and 13.60 min. The system ran with 20% A until 17 min for column cleaning. Finally, the mobile phase composition returned to the initial 99% A between 17.60 and 18.10 min and was maintained until 21.10 min for column re-equilibration before the next injection.

For each analysis, a 10 μL sample volume of the dry extract solution was injected, and the samples were stored at 8 °C in the autosampler. Quantification was performed using the PDA detector, with absorbance monitored at 265 nm for protocatechuic acid, vanillic acid, and syringic acid, and at 325 nm for chlorogenic acid, caffeic acid, p-coumaric acid, and ferulic acid. The QDa detector was used solely for mass confirmation of the eluted compounds by targeting their specific mass-to-charge ratios (*m*/*z*) in negative ion mode, such as 153 for protocatechuic acid, 163 for p-coumaric acid, 167 for vanillic acid, 179 for caffeic acid, 193 for ferulic acid, 197 for syringic acid, and 353 for chlorogenic acid.

Quantification relied on external standards, including protocatechuic acid, chlorogenic acid, vanillic acid, caffeic acid, syringic acid, p-coumaric acid, and ferulic acid. Stock solutions of these standards were prepared in methanol at a concentration of 1 mg/mL and serially diluted to create calibration curves ranging from 0.1 μg/mL to 50 μg/mL, which were analyzed using the PDA detector signals.

### 2.11. Acute Toxicity Assay of Extracts on the Species Daphnia Magna

For the assessment of acute toxicity on invertebrates (planktonic crustaceans), the experimental standard procedure was used as described in SR EN ISO 6341:2013/OECD 202—Water quality. Determination of mobility inhibition for *Daphnia magna* Straus (*Cladocera*, *Crustaceans*). Acute toxicity test (Daphtoxkit F magna kit—MicroBiotests Inc., Ghent, Belgium).

The method involves exposing the planktonic crustaceans (*Daphnia magna*) to different concentrations of the assessed sample in plates that allow testing in replicates. The effects on the tested organisms are monitored for a period of 24 to 48 h. The response is evaluated as a function of exposure concentration, compared to the effects of mortality or immobilization observed in the control samples. The concentration that causes a 50% mortality or immobilization effect in the test batch (EC50) is determined from the recorded mortality percentages across the series of wastewater sample concentrations.

After incubating the organisms for 48 h at 20–22 °C and 6000 lux in a suitable culture medium, the neonates were obtained and subjected to the toxic action of the plant extracts for 48 h. In accordance with the standard procedure method, the sample is tested at least 5 different concentrations, which ensures the accurate identification of the concentration at which no effect on the assessed organisms is observed, as well as the concentration at which the toxicity effect exceeds 50% compared to the control test. The samples of plant extracts: *U. dioica* (UDSE), *M.* × *piperita* (MESE), *C. oficinalis* (COSE), *C. oficinalis* + *J. regia* (CJSE)—were tested at the following concentrations of 2000, 1500, 1000, 500, 250, 125, 62.5, 62.5, 31.25, 15, 7.81 mg/L. These dilutions were prepared from individual stock solutions of 1000 mg/L in nutrient medium using an ultrasonic water bath (Elmasonic 1002238-C, ELMA, Singen, Germany) at a temperature of 30 °C for approximately 20 min. For statistical evaluation, each assessment was performed in triplicate, along with a control test that contained no plant extract. The samples were incubated in the dark at 20 °C in the Aqualytic Incubator TC 135S, Dortmund, Germany. The results were recorded after 24 h and 48 h of incubation. During incubation, the pH and oxygen parameters were monitored for each sample. The experimental data were interpreted using the following equation. For data evaluation, the Excel program MBT Daphnia Regtox 2.0 (1991, MicroBioTests Inc., Ghent, Belgium)—HILL model, provided by MicroBioTests Inc., was employed.$$\frac{no.\;dead\;organisms+no.\;immobile\;organisms}{20}\times 100=\%\;effect$$

In addition, the data was validated by the potassium bichromate assay (EC50 48 h = 0.92 mg/L) in conformity with the batch certificate DM 160524 (MicroBioTests Inc., Ghent, Belgium).

### 2.12. Phytotoxicity Assay

The evaluation of phytotoxicity effects of plant extracts was performed using the PHYTOTOTOXKIT microbioassay, by applying the methodology described in the international standard ISO 11269-1:2013—Determination of the impact of pollutants on soil—Part 1: Method for measuring root growth.

The test consists of the direct contact of the contaminated soil sample with 3 types of terrestrial plant seeds: *Sinapis alba* (dicotyledonous), *Lepidium sativum* (dicotyledonous), and Sorghum saccharatum (monocotyledonous), and the evaluation of inhibition of seed germination and early root growth compared to the control (reference soil). The reference soil is an artificial soil with the composition indicated in the PHYTOTOTOXKIT microbioassay: 85% sand, 10% kaolin, 5% peat (MicroBioTest Inc., Belgium).

The seeds of the selected plants were placed in the test plate at equal distances of each other. A black filter paper was placed on top of the sample and was fully hydrated with distilled water. After closing the test plate, it was incubated for 3 days at 25 °C in the dark. Tests were conducted in double replicates to ensure statistically significant results. After the testing period, the plates were photographed one at a time, and the images were analyzed using ImageJ, version 1.54k.

In accordance with the applied methodology, control tests (using uncontaminated reference soil) and reference tests (using boric acid as the reference substance) were conducted in parallel with the test samples. Each test was performed in 2 replicates.

The following parameters were calculated with the accumulated data: Seed germination inhibition and Root growth inhibition, using the equation below:$$I\%\;=\;A\;-\;\frac{B}{A}\;\times \;100$$
where A—average number of germinated seeds in the control sample or average root length in the control sample, B—average number of germinated seeds in the tested sample or average root length in the tested sample.

### 2.13. Statistical Analysis

All experiments were carried out with at least five replicates, and the data was presented as the mean ± standard deviation (SD). Data analysis was performed using XLSTAT Premium Software v2025.11.1429 (Lumivero, Denver, CO, USA). The one-way ANOVA test identifies statistically significant differences between variables (*p* < 0.05). Principal Component Analysis using Pearson correlation was selected to investigate the correlations between bioactive compounds and the pharmacological effects of plant extracts.

## 3. Results

### 3.1. Total Polyphenol Content (TPC)

Following the TPC evaluation, the data were presented in Table 1. To be noted, the data obtained for the dry plant extracts is expressed as mg tannic acid Eq./g of dry extract.

What can be concluded from this set of data, is that the extracts from *M.* × *piperita* are certainly more abundant in these phytoconstituents, the best method of extraction from a yield point of view being the dry extract procedure (this applies to all the extracts). Although the dry extract procedure is exceedingly effective at concentrating the polyphenols, it is necessary to note that the dry extract represents a highly concentrated form of active compounds from the extractive solution. In addition, the extraction which used as a solvent methanol 50% solution managed to concentrate more polyphenols than the extraction which used methanol 70% solution. This finding can be applied for the other plant extracts too, considering the most effective way of concentrating this class of phytocompounds, other than the dry extract procedure, is the procedure which uses 50% methanol solution as a solvent. From another point of view, the rationale for using these procedures, which involves the use of 50% methanol solution as solvent, may be related to cost-effectiveness, using much less reagent than the other method.

Following the *M.* × *piperita*, the extracts from the combination of *C. officinalis* and *J. regia* leaves present a relatively high number of compounds. Whether the total amount of polyphenols in each species, or other processes and mechanisms facilitating extraction, led to the concentration of more polyphenolic compounds, is difficult to determine; further studies are necessary to understand the process in more detail.

The data from the literature indicate a relative difference between the different TPCs of all the plant extracts. It may depend on the pedoclimatic factors or the various extraction techniques. H.J. Damien Dorman et al. [24] note that the extract from *M.* × *piperita* contains 262.8 ± 8.5 mg Galic acid Eq./g of plant material; however, the extraction process was drastically different. According to the approach presented in this paper, the extraction process involved a Soxhlet apparatus, and a sequential extraction was performed using a mixture of various solvents, including methanol. Sabina Begić et al. [25] state that *U. dioica* extracts contain 12.93 mg gallic acid eq./g plant material, which is roughly in line with our paper’s finding. However, the paper employs a different extraction process, specifically ultrasound-assisted extraction with pure methanol as the solvent. Regarding the mixture of *C. officinalis* and *J. regia* leaves, as of the research for this paper, no studies have been conducted that document the TPC of this particular mixture. However, Butnariu M. et al. [26] states that the maceration process of two *C. officinalis* varieties extracts, more precisely a maceration extraction with an 80% methanol solution, contain 134 mg of Galic acid Eq./100 mL of extractive solution, respectively, 153 mg of Galic acid Eq./100 mL of extractive solution, which is relatively close to this paper’s data.

### 3.2. Total Flavonoid Content (TFC)

After the TFC evaluation, the data are presented in Table 2. It is worth noting that the results for the dry plant extracts are expressed as mg Rutoside equivalent per 100 g of dry extract.

Flavonoids are known to possess antimicrobial properties by interacting with extracellular soluble proteins and bacterial cell wall components. Using these bio-treatments, which contain this class of compounds along with others (tannins, polyphenols, and phenolic acids), may lower the possibility of plant pathogenic infections during the stages of plant growth when the individual is prone to infections.

It appears that the highest concentration of flavonoids in all the extracts is found in ME50, indicating that the most effective method of flavonoid extraction for *M.* × *piperita* is the use of 50% methanol as the extraction solvent. Moreover, it is stated that increasing the concentration of methanol in the solvent negatively impacts the extraction process by decreasing the TFC. This effectiveness is also evident in the case of *C. officinalis* and *J. regia* leaf extractive solutions, with CJ50 categorically having a higher TFC than CJ70. It is also worth noting that, in the case of the CJM, the TFC is higher than that of the CJ70.

By using a different extraction technique (ultrasound-assisted extraction) and a different solvent (methanol 80% solution), M. Atanassova et al. [27] affirms that the *M.* × *piperita* has a TFC of 0.2517 mg Catechin Eq./g of plant material, which may further prove that the optimal condition to a higher concentration of TFC is using a solution of 50% methanol as solvent. However, the difference between the environmental conditions of the plant material sample examined in this study and the different approach of evaluating the TFC may be the answer to this notable difference.

Tarasevičienė Živilė et al. [28] has documented that the *U. dioica* leaves maceration using a solution of 50% methanol as solvent, contains a total of 5.37 mg of Quercetin Eq./g of dried plant material, and a maceration using a solution of 70% methanol as solvent, includes a total of 9.06 mg of Quercetin Eq./g of dried plant material. This set of data supports the conclusion that the concentration of the solvent is directly proportional to the concentration of TFC for this plant extract. However, further analysis is needed to confirm this affirmation. Still, there is a noteworthy difference of a few units of mg between this paper’s finding and the maceration finding, which may be explained by the different plant material used for extraction (leaves versus whole aerial parts).

As for the mixture of *C. officinalis* and *J. regia* leaves extracts, similar to the case of TPC, the TFC has not yet been documented, even though Butnariu M. et al. [26] managed to quantify the TFC for the two *C. officinalis* varieties as follows: 96.17 mg of Quercetin Eq./100 mL of solution (macerate) and 90.37 mg of Quercetin Eq./100 mL of solution (macerate). In contrast to the TPC, the difference between this paper’s findings and ours is of considerable size; it can be presumed that this class of natural compounds behaves very differently between maceration processes, depending on the solvent used or the different plant varieties analyzed. The *J. regia* extracts have been documented by Chaleshtori R. S. et al. [29] to contain a TFC of 270 ± 22.33 mg Rutoside Eq./g which is an indication that probably flavonoids from this particular species might concentrate better in an extract if ethanol is used as a solvent, or the pedoclimatic factors of the environment of which the plants sampled (Bakhtiari Province, Iran) in the study mentioned above, may drastically change the TFC.

### 3.3. Total Phenyl Carboxylic Acids Content (TAC)

Following the TAC evaluation, the data are summarized in Table 3. Notably, the results for the dry plant extract are expressed as μg of chlorogenic acid equivalent per 100 g of dry extract.

In relation to this class of compounds, specifically the phenolic acids, a decrease in the concentration of these active compounds in the extracts can be noted. One reason could be the sensitivity of this colorimetric technique of quantification, because Arnow’s reagent has a significantly increased sensitivity to react only with phenolic acids from this sizable phenolic class of plant compounds. Another reason might be that the chosen method of excision, or the solvent, does not efficiently manage concentrated phenolic acids as other methods and/or solvents do.

Jurić T. et al. [30] through an extraction process using 70% ethanol solutions manages to document that the TAC of *M.* × *piperita* plant material, commercially available in Gorenje, Slovenia, is 16.43 ± 0.75 mg of Caffeic acid Eq./g of dry weight of plant material, further adding on the assumption that phenolic acids may extract with a higher effectiveness if ethanol is used as a solvent. In the case of *U. dioica*, the TAC has been documented as 1.2686 ± 0.0586% g Chlorogenic Acid Eq., which supports the assumption made above about the efficacy of the methanol solvent being lower. Although the extraction process in this study involved methanol, it employed a sequential extraction technique [31]. It is also worth mentioning that there has been documentation indicating that male individuals of this species contain a higher concentration of TAC than female individuals, and that the highest percentage of phenolic acids occurs just before the blooming period [32].

CJ50 and CJ70 showed lower values than those documented in the scientific literature, although for J. regia, the TAC of the extract from leaves has not yet been demonstrated, only from the fruit husk or inflorescence [33,34].

### 3.4. Radical Scavenging Activity—ABTS Method

Following the compilation of the antioxidant activity data of extracts, Table 4 was generated.

It has been well established for a considerable period that the phenolic class of compounds exhibits a remarkable antioxidant and free radical scavenging capacity [35]. Therefore, the antioxidant and free radical scavenging activity that these three extracts present can be justified by the content of phenolic compounds (flavonoids, polyphenols, phenolic acids, etc.), previously documented in the earlier chapters.

Remarkably, the dry extract from *U. dioica* had a superior IC50 value than the other two extracts. Despite this paper’s finding, this species has an overall lower concentration of phenolic compounds. The scientific literature documents that the protein fraction of *U. dioica* extracts has an IC50 of 0.0199 mg/mL, which is almost three times lower than the value of this paper’s finding [36]; however, the antioxidant activity of proteins is lower than that of phenolic compounds, due to their chemical structure. Nonetheless, there is documentation that states proteins and phenolic compounds can interact to boost their overall antioxidant activity [37], which may be the case with this plant extract. However, the nature of the solvent used does not allow for a high concentration of proteins in the final extract.

Although *M.* × *piperita* seemed to have a promising antioxidant activity, due to its high concentration of phenolic compounds, the IC50 of MESE was relatively lower than the IC50 of UDSE. The IC50 of an extract (macerate) of *M*. × *piperita* was reported as 153.80 μg/mL [38], which is a substantial difference between this value and the one in this paper’s one; still, for the extraction process, quite a sizeable amount of plant matter was used (100 g), which directly influenced the high concentration of natural compounds.

Variation in IC50 data in the scientific literature is also present for *C. officinalis* and *J. regia*; indeed, there is no documentation of the antioxidant capacity for this plant mixture. Preethi K. C. et al. [39] reported that the IC50 of an Indian grown *C. officinalis* is 0.0065 mg/mL, and Rahata D. A. et al. [40] reported an IC50 equal to 0.310 mg/mL of the leaf extract of *J. regia*, further confirming variation in the scientific literature.

### 3.5. Diphenyl-1-Picrylhydrazyl Free Radical Scavenging Assay (DPPH)

Following the compilation of the DPPH antioxidant activity data of extracts, Table 5 was generated.

The DPPH radical scavenging assay demonstrated apparent differences in antioxidant potential among the tested extracts. MESE exhibited the highest activity with an IC50 of 0.1274 mg/mL, consistent with earlier reports showing vigorous radical scavenging activity of peppermint extracts, with IC50 values typically below 0.3 mg/mL depending on solvent and extraction method [41,42]. UDSE displayed moderate antioxidant activity (IC50 = 0.5475 mg/mL), comparable to values previously reported, where nettle leaf extracts showed IC50 values around 0.3 mg/mL [43]. In contrast, CJSE exhibited weaker activity (IC50 = 0.5931 mg/mL). This result aligns with prior findings, as *C. officinalis* extracts often exhibit IC50 values greater than 0.55 mg/mL [44], whereas *J. regia* leaves typically present reduced activity [45]. The reduced efficiency of the combined extract suggests potential antagonistic effects between the phenolics of *Calendula* and *Juglans*. Overall, our data reinforce the superior antioxidant potential of *M.* × *piperita* compared with the other studied extracts.

### 3.6. Ferric-Reducing Antioxidant Power Assay (FRAP)

Following the compilation of the FRAP assay data of extracts, Table 6 was generated.

MESE exhibited the strongest reducing capacity (EC50 = 0.1520 mg/mL), consistent with previous reports highlighting peppermint’s rich phenolic composition and high ferric ion-reducing ability [46,47]. UDSE demonstrated lower activity (EC50 = 0.4735 mg/mL), which aligns with published data showing nettle extracts possess moderate FRAP values relative to other medicinal plants [48,49]. CJSE showed intermediate potency (EC50 = 0.2639 mg/mL). This outcome is coherent with findings that *C. officinalis* flowers generally yield modest FRAP values [50,51], while *J. regia* leaves and kernels exhibit robust reducing capacities due to their abundance of ellagitannins and flavonoids [52,53,54]. These results suggest that peppermint extract is the most potent antioxidant in terms of ferric-reducing activity, while the walnut component in the mixture substantially contributes to its higher activity compared to *C. officinalis* alone.

### 3.7. Microelement Assay

Microelements play a very critical role in plant life. They contribute to the biochemical processes of plant metabolism, serving as cofactors for a multitude of enzymes (Fe, Cu, etc.). Phosphorus is part of the DNA structure, and magnesium is of crucial importance in the photosynthesis process, as it is a component of the porphyrin molecules that comprise chlorophyll. The lack of them in the soil leads to problems of the development and normal functioning of plant metabolism.

In Table 7, the data have been compiled for some of the most relevant microelement concentrations found in the plant material used for the extraction processes presented in the previous chapters, as well as the microelement assay for some extracts.

It is evident that during the extraction process, a significant percentage of microelements was lost due to chemical interactions between the solvent and the microelements. The *C. officinalis* and *J. regia* leaves extracts appear to have the highest ratio of micronutrients among all the plant extracts; this is also valid for the plant material. Next, from a concentration point of view, would be *M.* × *piperita* extract, which has a higher content of manganese than the *C. officinalis* and *J. regia* leaves extract.

There has been documentation of high levels of magnesium, zinc, and manganese in *M.* × *pieprita,* which aligns with our findings [55]. This confirms not only that the plant material contained high levels of these three microelements, but also that the extractive solution had relatively high levels of them. Different levels of microelements have been recorded in *U. dioica*. In Serbia [56], three plant material samples from other locations have been analyzed, and the data vary from the dataset presented above. This discrepancy can be attributed to the variance in soil microelements; however, the levels of zinc from *U. dioica* plant matter found in Serbia are similar to those in our study.

The same assumption can be made about the micronutrient concentrations of *C. officinalis* and *J. regia*, given that the soil in different parts of the world contains varying quantities of trace elements, which directly influence plants [57]. Consequently, the scientific data varies significantly between documents.

### 3.8. In-Depth UHPLC-MS Polyphenol Assay

In addition to the TPC, a thorough UHPLC-MS assay was conducted to further expand and confirm the presence of certain types of polyphenolic compounds found in the dry extracts. The results of the analysis are presented in Table 8, along with the corresponding chromatograms (Figure 1, Figure 2 and Figure 3).

From this data, it is readily apparent that the three extracts differ in their concentration of specific polyphenols. For example, the *U. dioica* extract shows a high concentration of chlorogenic acid and p-coumaric acid. When it comes to the *M.* × *piperita* extract, there is a higher content of protocatechuic acid, chlorogenic acid (the highest content of all three extracts), caffeic acid, and ferulic acid. In contrast, the *C. officinalis* and *J. regia* extracts have the highest concentration of syringic acid and a noticeable amount of chlorogenic acid. This data is supported by the findings of previous studies present in the scientific literature, with slight variation due to the nature of natural products, which are influenced by pedoclimatic factors. For instance, Patrizia Pinelli et al. [58] found high concentrations of chlorogenic acid in both cultivated and wild *U. dioica*. The high content of chlorogenic acid from the *C. officinalis* has been documented from a hydro-macerate [59].

These compounds have been correlated with insecticidal properties. Eldesouky S.E. et al. [60] link the presence of syringic acid and caffeic acid, alongside other polyphenolic compounds from *U. dioica* extracts, to the insecticidal effect on *Aphis gossypii* Glover and *Phenacoccus solenopsis* Tinsley and acetylcholinesterase and glutathione-S-transferase inhibition potential. Furthermore, the tannins extracted from *U. dioica* leaves have been documented to have the highest antifeedant activity and induced notable larval growth inhibition on *Spodoptera littoralis* (Boisduval), from tannins extracted from four natural products (*U. dioica* leaves, bean hulls, black tea leaves, and green tea leaves) [61]. In a comparative study of polyphenols extracted from *M.* × *piperita* and Ricinus communis, it has been documented that the polyphenols extracted from *M.* × *piperita* exhibit increased insecticidal activity against *Aphis spiraecola* P. compared to those from *R. communis*. Some of the polyphenols found in the extracts include syringic acid and chlorogenic acid.

There has not yet been any correlation between the polyphenols found in the C. officinalis extracts and the insecticidal properties of said extracts, even though the extract has been shown to have insecticidal and toxic properties on the species *Oncopeltus fasciatus* [62]. It has been mentioned in the scientific literature that the aqueous extract of the root bark of *J. regia* exhibits potential insecticidal properties, supported by the presence of compounds such as coumarin, epicatechin, and juglone; however, further scientific data are needed to substantiate this finding [63].

### 3.9. Acute Toxicity Assay of Extracts on the Species Daphnia Magna

In Table 9, the data obtained from this toxicity assay are presented, along with a graph for each of the extract assays. It is noted that the UDSE has an acute toxicity effect starting from >250 mg/L, with 100% mortality observed at 1000 mg/L. However, the extract is not considered toxic because its EC50 is greater than 100 mg/L, according to the toxicity classification of chemical substances. The same can be said about the other extracts as well, with slight differences, such as COSE and CJSE, which exhibit a concentration at which the 100% mortality rate appears at 500 mg/mL. Comparing the EC50 values obtained at 48 h for each of the extracts, COSE exhibits the highest toxicity, followed by CJSE and MESE, while UDSE has the lowest toxicity. This trend is also observed for the decreased exposure time (24 h), except for the *M.* × *piperita* extract, which does not exhibit significant toxic effects up to 500 mg/L (Figure 4).

From the practical usage point of view, the *C. officinalis* extract and the combination of *C. officinalis* and *J. regia* may indicate a biocide activity; however, further assays need to be performed to clarify and confirm the target mechanism of action.

The data available in the scientific literature suggest that extracts from *M.* × *piperita* (essential oil) exhibit mild toxicity to Daphnia magna. In contrast, aqueous extracts from *U. urens* and ethanolic extracts of *C. officinalis* have moderate to low toxicity. These plant extracts may be promising alternatives to synthetic biocides, but rigorous evaluation of their efficacy and environmental safety is essential [64,65,66,67,68]. Studies indicate that U. dioica extracts have low toxicity. For example, *U. dioica* extracts showed LC50 values greater than 1000 μg/mL in tests on Artemia salina, a model organism similar to *Daphnia magna*, suggesting low toxicity [69]. Data is limited for plant extracts in the lyophilized state, as well as for medicinal plant species used, particularly marigold and walnut.

### 3.10. Phytotoxicity Assay

The plant extracts evaluated neither produce any seed germination inhibition, nor any root growth inhibition on the chosen assessed plant species (*Sinapis alba*, *Lepidium sativum* and *Sorghum saccharatum*). The concentration range used for this test showed only 0 to 30% root growth inhibition which it is considered insignificant, but despite that, it has been recorded that the plant extract had a positive effect on root growth for *Sorghum saccharatum*, particularly the *M.* × *piperita* extract. According to the scientific literature, some aqueous plant extracts can enhance seed germination and root growth in plants, while others can inhibit these processes [70]. Nevertheless, no other studies have been identified to support and compare this data, leaving an open gap in information to be investigated.

The tests complied with the validity criteria of the method, i.e., in the control, the average seed germination was 100% for each of the plant species tested, and the average root length was greater than 30 mm for each species. The reference test revealed the sensitivity of the seedlings to toxic compounds (boric acid at 250 mg/L), with root growth inhibition ranging from 37% to 49% (Table 10 and Table 11 and Figure 5).

These plant extracts are widely used in various areas, particularly for human therapeutic purposes; however, there is no published experimental evidence of their direct effect on germination or root growth in other plants. Instead, their indirect benefits include stimulating biodiversity and soil health through the supply of nutrients and bioactive substances.

Recent studies show that extracts of *Salvia officinalis* and *Helianthus annuus* at a concentration of 100 g/L significantly inhibit the germination and metabolism of plants of economic interest, such as white *Sinapis sp.* and *Brassica rapa*. These extracts also cause membrane damage and disrupt seedling metabolic activity [71].

### 3.11. Discussions

The correlations between bioactive compounds and the pharmacological properties of the extracts were investigated using Principal Component Analysis (Figure 6). It reveals that the first two components account for all data variables (F1 = 62.99%, F2 = 37.01%, Figure 6A and F1 = 59.01%, F2 = 40.99%, Figure 6B), encompassing all the information in the dataset.

Protocatechuic acid, vanillic acid, caffeic acid, and fumaric acid are strongly correlated with EC50 (24 h) and *Sorghum saccharatum* germ size (r = 0.809–0.922, *p* > 0.05, Figure 6A). They also exhibit a moderate to robust negative correlation with DPPH-IC50 and FRAP-IC50 (r ranges from –0.773 to –0.997, *p* > 0.05, Figure 6A). Therefore, the increased content of these phenolic compounds, which exhibit considerable antiradical activity, induces *S. saccharatum* seed germination and diminishes the extracts’ cytotoxicity. Syringic acid highly correlates with *S. alba* germ size, and DPPH-IC50 (r = 0.955, r = 0.991, *p* > 0.05, Figure 6A), and chlorogenic acid with *L. sativum* germ size (r = 0.955, r = 0.901, *p* > 0.05, Figure 6A). In contrast, p-coumaric acid is considerably associated with EC50 (48 h) and ABTS-IC50 (r = 0.907, r = 0.906, *p* > 0.05, Figure 6A), and moderately correlates with EC50 (24 h) and *S. saccharatum* germ size (r = 0.784, r = 0.741, *p* > 0.05; Figure 6A). In contrast, chlorogenic acid considerably negatively correlates with EC50 (48 h) and ABTS–IC50 (r = −0.816, r = −0.817, *p* > 0.05, Figure 6A), while p-coumaric acid is negatively correlated with *S. alba* and *L. sativum* germ size (r = −0.816, r = −0.823, *p* > 0.05, Figure 6A). Appreciable negative correlations occur between TPC, vanillic acid, caffeic acid, ferulic acid, and *S. alba* root size (r = 0.805–0.869, *p* > 0.05, Figure 6A), suggesting their inhibitory effects on *S. alba* seed germination. Similarly, protocatechuic acid displays a moderate negative correlation with *S. alba* root size (r = 0.776, r = 0.753, *p* > 0.05, Figure 6A), while syringic acid shows a strong negative correlation with EC50 (24 h) and *S. saccharatum* germ size (r = −0.983, r = −0.969, *p* > 0.05, Figure 6A).

As expected, TPC and TFC displayed moderate to high negative correlations with FRAP-IC50 and DPPH-IC50 (r ranged between –0.783 and –0.994, *p* < 0.05, Figure 6B); therefore, they have shown a moderate to strong positive correlations with EC50 (24 h) and *S. saccharatum* germ size (r = 0.787–0.910, *p* > 0.05, Figure 6B). Minerals such as Ca, Mn, P, Cu, and Fe exhibit a strong negative correlation with ABTS-IC50 and EC50 (48 h) (r = 0.926–0.996, *p* > 0.05, Figure 6B), They also show a remarkable correlation with *L. sativum* germ size (r = 0.851–0.966, *p* > 0.05, Figure 6B) and a moderate negative correlation with EC50 (24 h) (r = 0.618–0.786, *p* > 0.05, Figure 6B), and a moderate correlation with *S. alba* root size (r = 0.575–0.751, *p* > 0.05, Figure 6B). However, Zn displays a high negative correlation with EC50 (24 h) (r = 0.987, *p* > 0.05, Figure 6B) and a strong positive correlation with *S. alba* root size (r = 0.977, *p* > 0.05, Figure 6B). Finally, Mn shows a strong negative correlation with FRAP-IC50 and DPPH-IC50 (r = −0.820, r = −0.985, *p* > 0.05, Figure 6B), as well as a moderate negative correlation with ABTS-IC50 (r = −0.506, *p* > 0.05, Figure 6B). As expected, it also displays a moderate correlation with EC50 (24 h) and with *L. sativum* and *S. saccharum* germ size (r = 0.642, r = 0.599, *p* > 0.05, Figure 6B).

A substantial positive correlation exists between the ABTS–IC50 and EC50 values measured after 48 h (r = 0.999, *p* < 0.05, Figure 6A,B). The EC50 measured after 24 h displays a significant positive correlation with *S. saccharum* root size (r = 0.998, *p* < 0.05) and a negative correlation with *S. alba* root size (r = 0.999, *p* < 0.05, Figure 6A,B). Moreover, DPPH-IC50 is highly correlated with S. alba germ size (r = 0.905, *p* > 0.05, Figure 6A,B).

All correlations are associated in a higher measure with MESE and CJSE than with UDSE (Figure 7). The Dendrogram from Figure 6C shows considerable dissimilarities between all extracts: 56% (40.32) between the two clusters (C1 and C2), and 44% (31.67) between MESE and CJSE (Figure 6C).

## 4. Conclusions

This research validates the efficacy of traditional Romanian ethnobotanical practices, particularly the use of plant-based bio-treatments, in promoting sustainable agriculture. The high concentrations of bioactive compounds, such as polyphenols, flavonoids, and phenolic acids, in the analyzed species underscore their potential as eco-friendly alternatives to conventional pesticides and fertilizers. Additionally, the observed microelement content supports soil fertility and plant health, further advocating for their integration into modern agricultural systems. Future studies should delve deeper into optimizing extraction techniques and understanding the synergistic effects of plant mixtures to maximize their agricultural and ecological benefits.

## Figures and Tables

**Figure 1 antioxidants-14-01198-f001:**
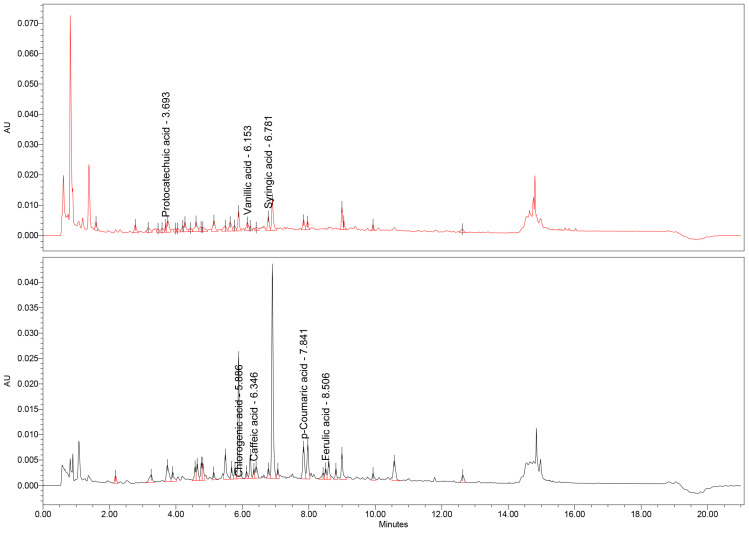
UHPLC-MS chromatogram of *U. dioica* extract.

**Figure 2 antioxidants-14-01198-f002:**
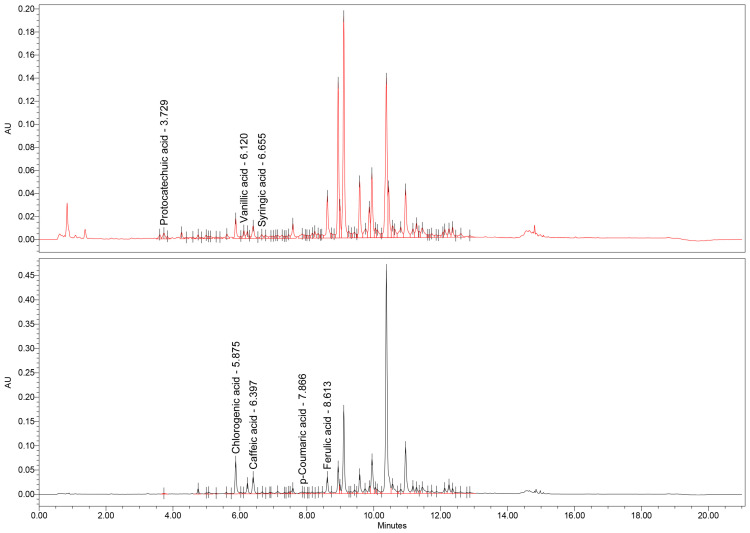
UHPLC-MS chromatogram of *M.* × *piperita* extract.

**Figure 3 antioxidants-14-01198-f003:**
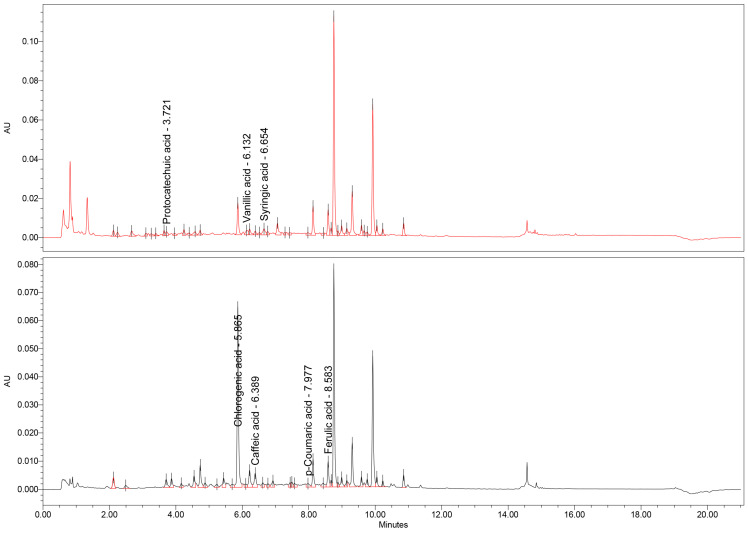
UHPLC-MS chromatogram of *C. officinalis* and *J. regia* extract.

**Figure 4 antioxidants-14-01198-f004:**
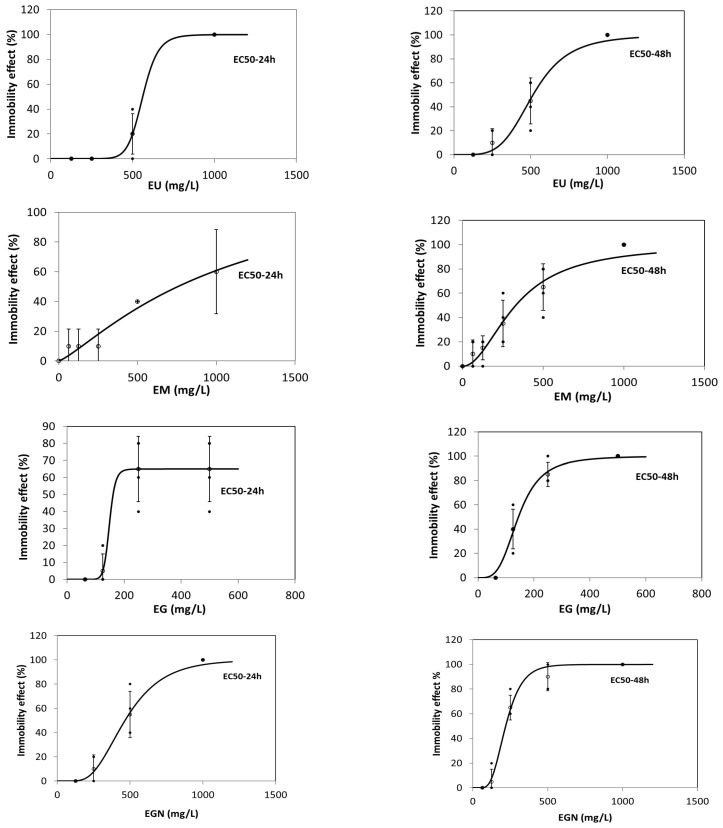
Graphs for the estimation of EC50 for 24 h and 48 h. EU—*U. dioica* solid extract; EM—*M.* × *piperita* solid extract; EG—*C. officinalis* solid extract; EGN—*C. officinalis* and *J. regia* extract.

**Figure 5 antioxidants-14-01198-f005:**
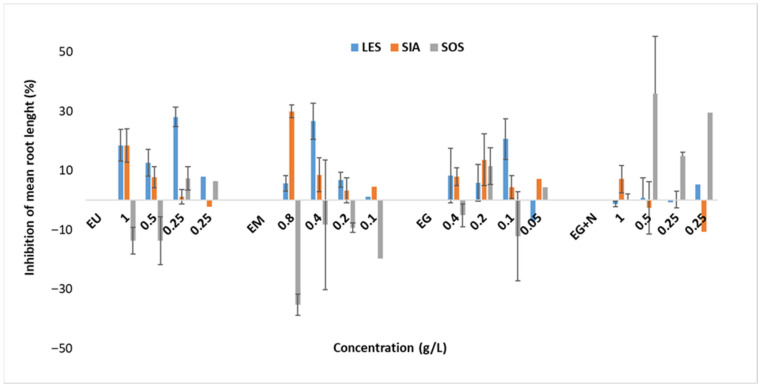
EU—*U. dioica* solid extract; EM—*M.* × *piperita* solid extract; EG—*C. officinalis* solid extract; EGN—*C. officinalis* and *J. regia* extract; LES—*Lepidium sativum*; SIA—*Sinapis alba*; SOS—*Sorghum saccharatum*.

**Figure 6 antioxidants-14-01198-f006:**
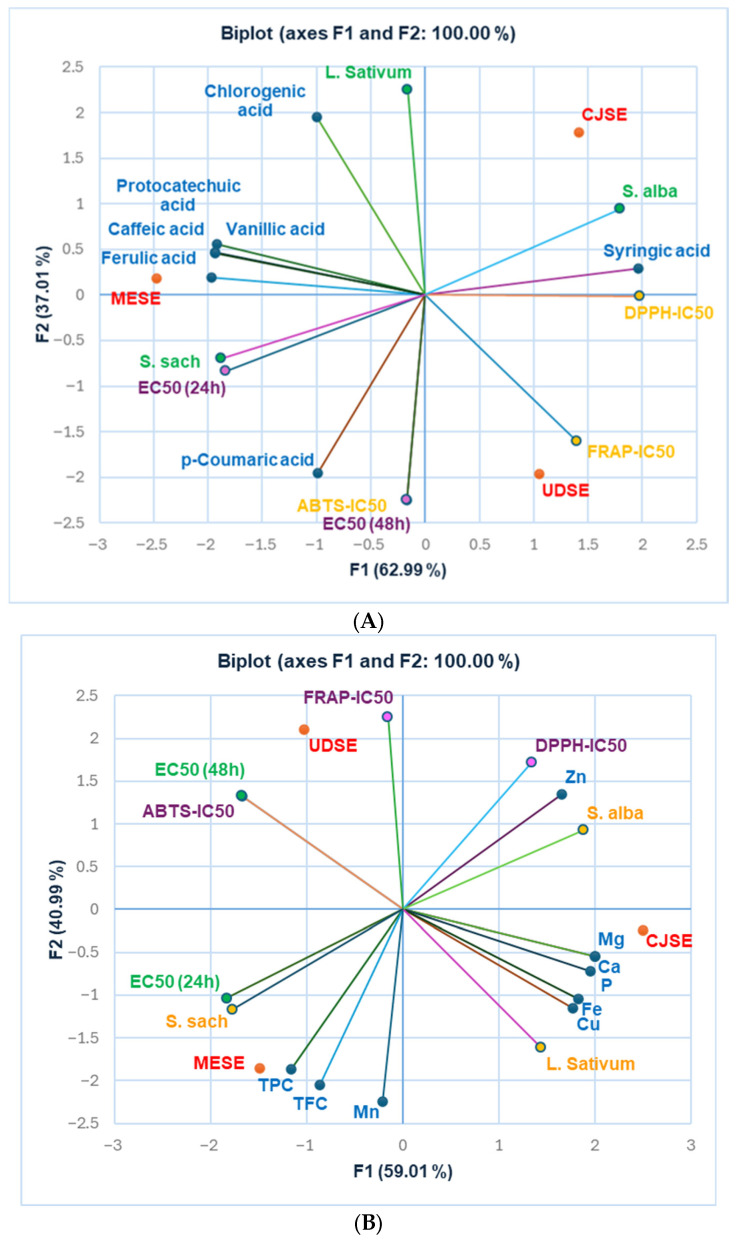
(**A**,**B**): Principal Component Analysis—Correlations between pharmacological properties of plant extracts and: (**A**) phenolic compounds identified by UHPLC-MS; (**B**) TPC, TFC, and minerals; (**C**) Agglomerative Hierarchical Clustering Dendrogram.

**Figure 7 antioxidants-14-01198-f007:**
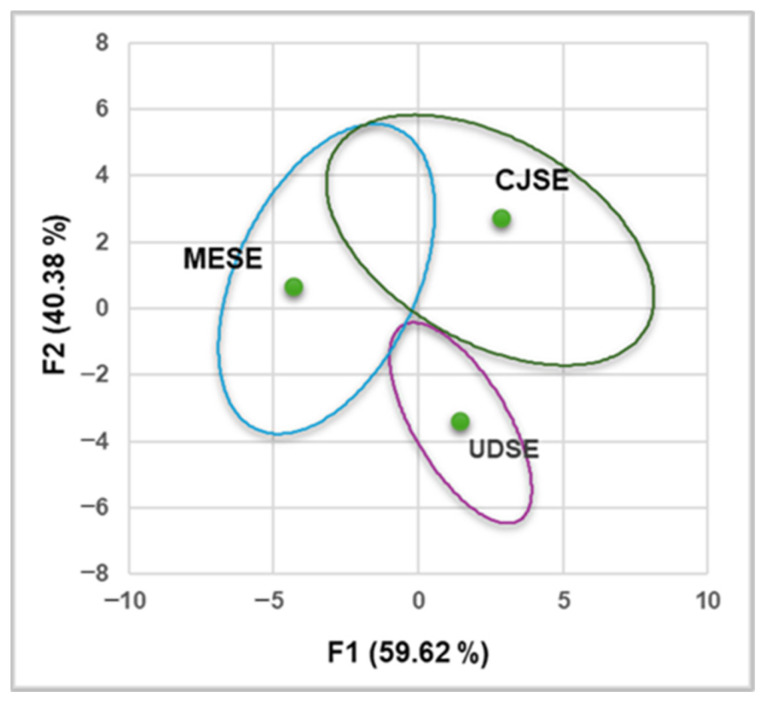
MESE, CJSE and UDSE correlations.

**Table 1 antioxidants-14-01198-t001:** The quantitative assessment of total polyphenol content in the plant extracts.

Plant Material	Plant Extract	TPC mg Tannic Acid Eq./g Plant Material
*M.* × *piperita*	MEI	34.9366 ± 0.4717 ^a,b,c^
	ME50	61.1526 ± 8.2845 ^a,b,c^
	ME70	57.9008 ± 4.2868 ^c^
	MESE	199.89 ± 4.7502 *^,a,b,c^
*U. dioica*	UD50	15.9209 ± 2.2722 ^d^
	UD70	9.4938 ± 1.0330 ^d^
	UDSE	42.1454 ± 1.7728 *^,d^
*C. officinalis* and *J. regia*	CJ50	21.3316 ± 2.1819 ^e^
	CJ70	18.0059 ± 1.0746 ^f^
	CJSE	47.5546 ± 1.8823 *^,e,f^

Total Polyphenol content (TPC). Results are expressed as mean ± SD (*n* ≥ 5). MEI—*M.* × *piperita* infusion extract; ME50—*M.* × *piperita* methanol 50% extract; ME70—*M.* × *piperita* methanol 70% extract; MESE—*M.* × *piperita* dry extract; UD50—*U. dioica* methanol 50% extract; UD70—*U. dioica* methanol 70% extract; UDSE—*U. dioica* dry extract; CJ50—*C. officinalis* and *J. regia* leaves methanol 50% extract; CJ70—*C. officinalis* and *J. regia* leaves dry extract. *—the data obtained for the dry plant extracts is expressed as mg Tannic acid Eq./100 g of dry extract; the values marked with the same superscript letter are significantly different (*p* < 0.05).

**Table 2 antioxidants-14-01198-t002:** The quantitative assessment of total flavonoid content in the plant extracts.

Plant Material	Plant Extract	TFC mg Rutoside Eq./100 g Plant Material
*M.* × *piperita*	MEI	10.2951 ± 0.2505 ^a^
	ME50	160.3046 ± 22.0657 ^a^
	ME70	17.0057 ± 1.7023 ^a^
	MESE	46.7516 ± 0.2597 *^,a^
*U. dioica*	UD50	1.7674 ± 0.2491 ^b^
	UD70	2.0794 ± 0.5615 ^c^
	UDSE	7.6281 ± 0.8178 *^,b,c^
*C. officinalis* and *J. regia*	CJM	1.8413 ± 0.0488 ^a^
	CJ50	4.4539 ± 0.2434 ^a^
	CJ70	0.1612 ± 0.0070 ^a^
	CJSE	15.8723 ± 0.9008 *^,a^

Total Flavonoid content (TFC). Results are expressed as mean ± SD (*n* ≥ 5). MEI—*M.* × *piperita* infusion extract; ME50—*M.* × *piperita* methanol 50% extract; ME70—*M.* × *piperita* methanol 70% extract; MESE—*M.* × *piperita* dry extract; UD50—*U. dioica* methanol 50% extract; UD70—*U. dioica* methanol 70% extract; UDSE—*U. dioica* dry extract; CJM—*C. officinalis* and *J. regia* leaves maceration extraction; CJ50—*C. officinalis* and *J. regia* leaves methanol 50% extract; CJ70—*C. officinalis* and *J. regia* leaves methanol 70% extract; CJSE—*C. officinalis* and *J. regia* leaves dry extract. *—the data obtained for the dry plant extracts is expressed as mg Rutoside acid Eq./100 g of dry extract; the values marked with the same superscript letter are significantly different (*p* < 0.05).

**Table 3 antioxidants-14-01198-t003:** The quantitative assessment of total phenyl carboxylic acids content in the plant extracts.

Plant Material	Plant Extract	TPC μg Chlorogenic Acid Eq./100 g Plant Material
*M.* × *piperita*	MESE	79.3036 ± 6.4262 *^,a,b,c,d^
*U. dioica*	UD50	3.00768 ± 0.2566 ^a,b,d^
	UD70	3.2136 ± 0.4137 ^b,c^
*C. officinalis* and *J. regia*	CJ50	6.0781 ± 0.6597 ^a,c,d^
	CJ70	5.2850 ± 0.5578 ^b^

Total Phenyl Carboxylic acids content (TAC). Results are expressed as mean ± SD (*n* ≥ 5). MESE—*M.* × *piperita* dry extract; UD50—*U. dioica* methanol 50% extract; UD70—*U. dioica* methanol 70% extract; CJ50—*C. officinalis* and *J. regia* leaves methanol 50% extract; CJ70—*C. officinalis* and *J. regia* leaves methanol 70% extract. *—The data obtained for the dry plant extracts is expressed as μg Chlorogenic acid Eq./100 g of dry extract; the values marked with the same superscript letter are significantly different (*p* < 0.05).

**Table 4 antioxidants-14-01198-t004:** ABTS free radical scavenging activity expressed as ABTS Inhibition % (IC) and IC50 for each of the dry extracts.

Plant Material	Plant Extract	ABTS Inhibition % (IC)	IC50 (mg of Dry Extract/mL)
*M.* × *piperita*	MESE	47.6369	0.0205
		56.0884	
		66.5138	
		70.9480	
		78.0790	
		82.8329	
		89.4495	
		92.2852	
		93.4529	
		99.1799	
*U. dioica*	UDSE	46.4971	0.0365
		55.0459	
		59.2160	
		74.6455	
		75.3684	
		81.3317	
		91.5207	
		93.5085	
		93.9533	
		98.1512	
*C. officinalis* and *J. regia*	CJSE	45.0097	0.0179
		58.0206	
		62.3158	
		72.6856	
		89.0742	
		91.4929	
		94.2313	
		96.2052	
		97.4423	
		98.6378	

ABTS free radical scavenging activity. Results are expressed by using the formula described above. IC50—Half maximal inhibitory concentration, expressed as mg of dry extract/mL of dry extract solution. MESE—*M.* × *piperita* dry extract; UDSE—*U. dioica* dry extract; CJSE—*C. officinalis* and *J. regia* leaves dry extract.

**Table 5 antioxidants-14-01198-t005:** DPPH radical scavenging assay expressed as DPPH Inhibition % (IC) and IC50 for each of the dry extracts.

Plant Material	Plant Extract	DPPH Inhibition % (IC)	IC50 (mg of Dry Extract/mL)
*M.* × *piperita*	MESE	31.1350	0.1274
		38.5915	
		49.3102	
		55.3072	
		68.3438	
		76.1160	
		83.5449	
		89.0177	
		90.4832	
		99.6382	
*U. dioica*	UDSE	15.7039	0.5475
		17.2799	
		19.7112	
		24.8344	
		27.8207	
		29.8228	
		32.0242	
		34.1029	
		37.1168	
		39.8240	
*C. officinalis* and *J. regia*	CJSE	12.5000	0.5931
		15.1245	
		16.6759	
		19.8093	
		21.9340	
		24.0005	
		26.7599	
		30.6843	
		35.6696	
		38.6191	

DPPH radical scavenging assay. Results are expressed by using the formula expressed above. IC50—Half maximal inhibitory concentration, expressed as mg of dry extract/mL of dry extract solution. MESE—*M.* × *piperita* dry extract; UDSE—*U. dioica* dry extract; CJSE—*C. officinalis* and *J. regia* leaves dry extract.

**Table 6 antioxidants-14-01198-t006:** FRAP assay expressed as IC50 for each of the dry extracts.

Plant Material	Plant Extract	EC50 (mg of Dry Extract/mL)
*M.* × *piperita*	MESE	0.1520 ± 0.3732 ^a^
*U. dioica*	UDSE	0.4735 ± 0.1514 ^a^
*C. officinalis* and *J. regia*	CJSE	0.2639 ± 0.1689 ^a^

FRAP assay. Results are expressed as EC50 value ± STDEV. EC50—Half maximal inhibitory concentration, expressed as mg of dry extract/mL of dry extract solution. MESE—*M.* × *piperita* dry extract; UDSE—*U. dioica* dry extract; CJSE—*C. officinalis* and *J. regia* leaves dry extract; the values marked with the same superscript letter in the same column are significantly different (*p* < 0.05).

**Table 7 antioxidants-14-01198-t007:** Quantitative assay of microelements.

Plant Sample	Ca (mg/kg)	Mg (mg/kg)	P (mg/kg)	Cu (mg/kg)	Fe (mg/kg)	Mn (mg/kg)	Zn (mg/kg)
*M.* × *piperita*—Plant material	560.2 ± 0.2502	585 ± 1.497	58.2 ± 0.378	14.92 ± 0.990	839.67 ± 59.56	107.89 ± 11.88	30.63 ± 3.995
*U. dioica*—Plant material	435.1 ± 0.5859	515 ± 1.928	60.6 ± 0.873	6.703 ± 0.688	207.37 ± 22.71	29.24 ± 2.487	30.18 ± 1.480
*C. officinalis* and *J. regia*—Plant material	903.8 ± 0.3785	1120.3 ± 0.608	88.3 ± 0.208	14.46 ± 0.594	880.75 ± 71.08	39.19 ± 2.033	34.30 ± 0.981
	**Ca (mg/L)**	**Mg (mg/L)**	**P (mg/L)**	**Cu (mg/L)**	**Fe (mg/L)**	**Mn (mg/L)**	**Zn (mg/L)**
*M.* × *piperita*—Extract	1.438 ± 0.006	1.554 ± 0.001	0.485 ± 0.065	0.156 ± 0.0106	0.659 ± 0.0453	1.601 ± 0.0252	0.348 ± 0.0451
*U. dioica*—Extract	1.277 ± 0.023	1.358 ±0.003	0.353 ± 0.003	0.071 ± 0.0040	0.277 ± 0.0300	0.100 ± 0.0100	0.732 ± 0.0361
*C. officinalis* and *J. regia*—Extract	2.368 ± 0.006	2.687 ± 0.001	0.938 ± 0.009	0.274 ± 0.0451	1.308 ± 0.0404	0.859 ± 0.0431	0.900 ± 0.0410

Results are expressed as mean ± SD (*n* ≥ 3). Results are expressed as mg of microelement/kg of plant material and mg of microelement/L of extract solution.

**Table 8 antioxidants-14-01198-t008:** Polyphenols quantified by UHPLC-MS.

Compound	*U. dioica* (mg/g Extract)	*M.* × *piperita* (mg/g Extract)	*C. officinalis* and *J. regia* (mg/g Extract)
Protocatechuic acid	0.276 ± 0.008 ^a^	1.417 ± 0.043 ^a^	0.476 ± 0.014 ^a^
Vanillic acid	0.159 ± 0.005 ^a^	0.371 ± 0.011 ^a,b^	0.159 ± 0.005 ^b^
Syringic acid	0.268 ± 0.008 ^a^	0.216 ± 0.006 ^a,b^	0.283 ± 0.008 ^b^
Chlorogenic acid	1.445 ± 0.043 ^a^	4.455 ± 0.134 ^a^	4.152 ± 0.125 ^a^
Caffeic acid	0.008 ± 0.000 ^a^	2.210 ± 0.066 ^a^	0.300 ± 0.009 ^a^
p-Coumaric acid	0.769 ± 0.023 ^a^	0.716 ± 0.021 ^b^	0.215 ± 0.006 ^a,b^
Ferulic acid	0.064 ± 0.002 ^a^	1.074 ± 0.032 ^a^	0.194 ± 0.006 ^a^

The values marked with the same superscript letter in the same row are significantly different (*p* < 0.05).

**Table 9 antioxidants-14-01198-t009:** The acute toxicity assay of extracts on the species *Daphnia magna*.

Extract	Concentration mg/L	Mortality/Immobilization %		EC_50_(24 h) mg/L *	EC_50_(48 h) mg/L *
		24 h	48 h		
UDSE	2000	100	100	561.63 ^a^ (528.40–602.23)	514.45 ^b^ (439.99–569.86)
	1500	100	100		
	1000	100	100		
	500	20	45		
	250	0	10		
	125	0	0		
	62.5	0	0		
	31.25	0	0		
	15	0	0		
	7.81	0	0		
	0	0	0		
MESE	2000	100	100	1070.47 ^a,x^ (256.48–4078.58)	329.61 ^b,x^ (266.20–404.83)
	1500	75	100		
	1000	60	100		
	500	40	65		
	250	10	35		
	125	10	15		
	62.5	10	10		
	31.25	0	0		
	15	0	0		
	7.81	0	0		
	0	0	0		
COSE	2000	100	100	148.37 ^a^ (135.31–226.62)	143.86 ^b^ (123.82–163.01)
	1500	100	100		
	1000	95	100		
	500	65	100		
	250	65	85		
	125	0	40		
	62.5	0	0		
	31.25	0	0		
	15	0	0		
	7.81	0	0		
	0	0	0		
CJSE	2000	100	100	464.70 ^a,x^ (398.50–536.19)	219.33 ^b,x^ (203.82–236.50)
	1500	100	100		
	1000	100	100		
	500	55	90		
	250	10	65		
	125	0	5		
	62.5	0	0		
	31.25	0	0		
	15	0	0		
	7.81	0	0		
	0	0	0		

* The value of CI 95% accompanies the data. UDSE—*U. dioica* solid extract; MESE—*M.* × *piperita* solid extract; COSE—*C. officinalis* solid extract; CJSE—*C. officinalis* and *J. regia* extract; the values marked with the same superscript letter (a and b in the same column and x in the same row) are significantly different (*p* < 0.05).

**Table 10 antioxidants-14-01198-t010:** Percentages of seed germination inhibition for plant extract samples.

Plant Extract	Concentration	Plant Species Assessed		
		*Lepidium sativum*	*Sinapis alba*	*Sorghum saccharatum*
*U. dioica*	1 g/L	10 0%	10 0%	10 0%
	0.5 g/L	10 0%	10 0%	10 0%
	0.25 g/L	10 0%	10 0%	10 0%
	0.125 g/L	10 0%	10 0%	10 0%
*M.* × *piperita*	0.8 g/L	10 0%	10 0%	10 0%
	0.4 g/L	10 0%	10 0%	10 0%
	0.2 g/L	10 0%	10 0%	10 0%
	0.1 g/L	9.5 5%	10 0%	10 0%
*C. officinalis*	0.4 g/L	10 0%	10 0%	9.5 5%
	0.2 g/L	10 0%	10 0%	10 0%
	0.1 g/L	9.5 5%	10 0%	10 0%
	0.05 g/L	10 0%	10 0%	9.5 5%
*C. officinalis* + *J. regia*	1 g/L	10 0%	10 0%	10 0%
	0.5 g/L	10 0%	10 0%	6.5 35%
	0.25 g/L	9.5 5%	10 0%	9.5 5%
	0.125 g/L	10 0%	10 0%	8 20%
Boric acid	250 mg/L	10 0%	10 0%	10 0%
Control		10 -	10 -	10 -

The data is expressed as the average number of germinated seed and germination inhibition %; *n* = 2 replicates.

**Table 11 antioxidants-14-01198-t011:** Percentages of inhibition of plant root growth for plant extract samples.

Plant Extract	Concentration	Plant Species Assessed		
		*Lepidium sativum*	*Sinapis alba*	*Sorghum saccharatum*
*U. dioica*	1 g/L	44.15 ± 1.07 mm 18.42 ± 1.97 ^a^ %	43.04 ± 2.69 mm 18.38 ± 5.09 ^b^ %	42.69 ± 2.47 mm −13.69 ± 6.59 ^a,b^ %
	0.5 g/L	47.35 ± 2.90 ^a^ mm 12.50 ± 5.35%	48.70 ± 2.96 ^b^ mm 7.65 ± 5.60%	34.81 ± 1.66 ^a,b^ mm 7.28 ± 4.43%
	0.25 g/L	38.94 ± 2.44 ^a^ mm 28.04 ± 4.51%	57.57 ± 1.90 ^a,b^ mm 1.11 ± 3.59%	35.13 ± 3.03 ^b^ mm 6.45 ± 8.07%
	0.125 g/L	49.86 ± 1.77 ^a^ mm 7.85 ± 3.27%	53.89 ± 1.27 ^a^ mm −2.18 ± 2.41%	25.82 ± 1.46 ^a^ mm 31.24 ± 3.90%
*M.* × *piperita*	0.8 g/L	51.06 ± 1.63 ^a^ mm 5.64 ± 3.00%	36.95 ± 1.70 ^a,b^ mm 29.93 ± 3.21%	50.82 ± 1.27 ^b^ mm −35.24 ± 3.98%
	0.4 g/L	39.73 ± 1.45 ^a^ mm 26.57 ± 2.68%	48.27 ± 1.14 ^a,b^ mm 8.48 ± 2.15%	40.67 ± 1.34 ^b^ mm −8.31 ± 3.57%
	0.2 g/L	50.43 ± 3.28 ^a^ mm 6.81 ± 6.06 ^a^ %	51.01 ± 3.01 ^b^ mm 3.27 ± 5.70 ^b^ %	41.05 ± 8.20 ^a,b^ mm −9.31 ± 21.82 ^a,b^ %
	0.1 g/L	53.54 ± 1.37 ^a^ mm 1.06 ± 2.53 ^a^ %	50.35 ± 2.25 ^b^ mm 4.45 ± 4.25 ^b^ %	44.93 ± 7.35 ^a,b^ mm −19.65 ± 19.58 ^a,b^ %
*C. officinalis*	0.4 g/L	49.68 ± 1.88 ^a^ mm 8.19 ± 3.47 ^a^ %	48.56 ± 1.48 ^b^ mm 7.91 ± 2.80%	39.48 ± 4.62 ^a,b^ mm −5.13 ± 12.30 ^a^ %
	0.2 g/L	50.94 ± 4.99 ^a^ mm 5.86 ± 9.22 ^a^ %	45.58 ± 1.60 ^b^ mm 13.57 ± 3.02 ^a^ %	33.27 ± 1.45 ^a,b^ mm 11.40 ± 3.85%
	0.1 g/L	41.90 ± 3.27 mm 20.56 ± 6.20 ^a^ %	51.73 ± 4.75 ^a^ mm 4.41 ± 8.77 ^a^ %	42.12 ± 2.32 ^a^ mm −12.17 ± 6.16 ^a^ %
	0.05 g/L	57.96 ± 3.75 ^a^ mm −7.10 ± 6.92 ^a^ %	49.01 ± 2.70 ^a^ mm 7.07 ± 3.91 ^a^ %	35.90 ± 5.66 a mm 4.39 ± 15.07 ^a^ %
*C. officinalis* + *J. regia*	1 g/L	54.85 ± 7.69 ^a^ mm −1.35 ± 14.21 ^a^ %	49.00 ± 2.42 ^b^ mm 7.09 ± 4.58 ^a^ %	37.47 ± 3.15 ^a,b^ mm 0.20 ± 8.39 ^a^ %
	0.5 g/L	53.66 ± 0.49 ^a^ mm 0.84 ± 0.91 ^a^ %	55.85 ± 2.43 ^b^ mm −2.63 ± 4.60 ^a^ %	24.12 ± 0.68 ^a,b^ mm 35.76 ±1.79 ^a^ %
	0.25 g/L	54.48 ± 3.64 ^a^ mm −0.67 ± 6.71 ^a^ %	52.63 ± 4.63 ^b^ mm 0.20 ± 8.78 ^a^ %	31.98 ± 7.28 ^a,b^ mm 14.83 ± 19.37 ^a^ %
	0.125 g/L	51.27 ± 1.33 ^a^ mm 5.26 ± 2.45 ^a^ %	58.35 ± 1.49 ^a^ mm −10.64 ± 2.83 ^a^ %	26.46 ± 5.2 ^a^ mm 29.50 ± 1.37 ^a^ %
Boric acid	250 mg/L	27.56 ± 2.96 ^a^ mm 49.05 ± 5.48 ^a^ %	30.56 ± 2.19 ^b^ mm 37.46 ± 2.33 ^a^ %	21.93 ± 2.98 ^a,b^ mm 41.58 ±7.95%
Control		10 ^a,b^ -	54.12 ± 4.52 ^a^ mm -	52.74 ± 4.30 ^b^ mm -

The data is expressed as the average root growth (mm) + Standard deviation and root growth inhibition %; *n* = 2 replicates; The values marked with the same superscript letter (a and b in the same row) are significantly different (*p* < 0.05).

## Data Availability

The datasets generated and analyzed during the current study are available from the corresponding author on reasonable request.

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
