# Peer review of "Phytochemical Composition and Antioxidant Activity of Traditional Plant Extracts with Biocidal Effects and Soil-Enhancing Potential"

_antioxidants, 2025, doi:10.3390/antiox14101198_

Round 1
Reviewer 1 Report
The limit of the study is the lack of antioxidant assays besides ABTS.
Abstract section: please check the plants' name. Please note that they have to be written in italics.
In materials and methods, please include a table with all details related to chromatographic separation.
Table 5: please include S.D. for each reported measurement.
Although the appreciable use of ecotoxicological models that permit to predict biocompatibility and (phyto)toxicity, the authors did not explore accurately the antioxidant potential of the tested extracts. This is a limit of the study, especially for a journal like Antioxidants.
It is suggestable to improve antioxidant evaluation during revision.
Please check the format of references.
Author Response
Dear Reviewer,
You can find attached the following document which goes into more details about the comments made previously.
Kind regards,
The authors

Reviewer 2 Report
Studies on biosustainable alternatives in agriculture are relevant, but some issues need to be better clarified and addressed by the Authors.
Some specific considerations:
1) I suggest that the "Keywords" be more specific and mention the extracts used in this study, such as "Calendula officinalis," "Mentha piperita," "Urtica dioica," and "Juglans regia." Please check the use of the term "Mentha × piperita";
2) Introduction: This section should provide a more concise and objective approach to the key points of the study, such as the characterization and properties of the different plant species investigated in this study (Calendula officinalis, Mentha piperita, Urtica dioica and Juglans regia), applications of their bioactive extracts, especially in agriculture, as well as a brief approach to extraction methods, incorporation techniques, association or not of species, and delivery systems for the bioactive material. Provide a broader overview of the literature on the biological properties of these plant extracts and their potential benefits for agriculture and sustainability. Organize the key points of the study better into interconnected paragraphs;
3) Materials and Methods: In Section 2.2, considering the factors that can influence the bioactive composition of plant extracts and impact their properties and functionalities, according to the literature, clarify and more clearly describe how the Authors defined the following issues regarding the design of this study: (A) the parts of the plant species used (lines 162-163); (B) the mixture of Calendula officinalis and Juglans regia species (and not the individual evaluation of each species); (C) the concentrations/ratios used; and (D) the extraction methods (including solvents) used in the study;
4) Results and Discussion: In the Tables and Figures, include data regarding Calendula officinalis and Juglans regia species separately, if applicable. Considering data available in the literature, discuss in more detail the aspects mentioned in comments 2 and 3, especially regarding the composition, properties and applicability of these species, especially in agriculture.
Author Response

(The authors gave the same response as above.)

Round 2
Reviewer 2 Report
The topic of the Authors' investigation is relevant.
Adjustments would be necessary to clarify the previous questions. I suggest that the Authors consider the previous comments more carefully.
Author Response
Dear Reviewer 2,
Thanks for making the great effort of revising the manuscript. We appreciated your opinions and helpful suggestions. We revised the current draft and we hope you’ll find it better organized, according to your appreciation.
Please find hereafter our response to your indications. Modifications have been shown in red color in the main text, using Track changes.
Best regards,
The authors
- Reviewer 1: As per previous comment, the "Introduction" section should provide a more concise and objective approach to the key points of the study, such as the characterization and properties of the different plant species investigated in this study (Calendula officinalis, Mentha piperita, Urtica dioica and Juglans regia), applications of their bioactive extracts, especially in agriculture, as well as a brief approach to extraction methods, incorporation techniques, association or not of species, and delivery systems for the bioactive material. Provide a broader overview of the literature on the biological properties of these plant extracts and their potential benefits for agriculture and sustainability. Organize the key points of the study better into interconnected paragraphs.
The authors:
Following feedback, the next additions were implemented lines: 47-96
- Reviewer 1: As per previous comment, in Section 2.2, considering the factors that can influence the bioactive composition of plant extracts and impact their properties and functionalities, according to the literature, clarify and more clearly describe how the Authors defined the following issues regarding the design of this study: (A) the parts of the plant species used (lines 162-163); (B) the mixture of Calendula officinalis and Juglans regia species (and not the individual evaluation of each species); (C) the concentrations/ratios used; and (D) the extraction methods (including solvents) used in the study.
The authors:
Following feedback, the next additions were implemented
A – regarding the plant parts used lines: 125-129 and 134-138
B – regarding the mixture of C. officinalis and J. regia lines: 138-142 and 184-189
C and D – regardin the concentrations and extraction methods lines: 98-115 and 143-157
- Reviewer 1: As per previous comment, in the Tables and Figures, include data regarding Calendula officinalis and Juglans regia species separately, if applicable. Considering data available in the literature, discuss in more detail the aspects mentioned in comments 2 and 3, especially regarding the composition, properties and applicability of these species, especially in agriculture.
The authors:
Following feedback, the next additions were implemented where the data was avilable.

Round 3
Reviewer 2 Report
The adjustments improved the manuscript.
The Authors have sufficiently responded to all previous comments and the manuscript has been satisfactorily adjusted.